# Mind the Gap: Professionalization is the Key to Strengthening Safety and Leadership in the Construction Sector

**DOI:** 10.3390/ijerph16112045

**Published:** 2019-06-10

**Authors:** Álvaro Romero, María de las Nieves González, María Segarra, Blasa María Villena, Ángel Rodríguez

**Affiliations:** 1Departamento de Construcciones Arquitectónicas y su Control, Universidad Politécnica de Madrid, 28007 Madrid, Spain; blasyvillena@hotmail.com; 2Department of Civil Engineering & Construction, University of Castile-La Mancha, 16071 Cuenca, Spain; Maria.Segarra@uclm.es; 3Department of Architectural Constructions & Construction Engineering and Land, University of Burgos, 09001 Burgos, Spain; arsaizmc@ubu.es

**Keywords:** training and information, professionalization, preventive culture, construction sector, OHS professionals, safety investments, safety management

## Abstract

The aim of this paper is to analyze the reality of risk prevention in construction sector companies in Spain, from the perspective of training, management, and risk prevention, as well as the amount of resources that are allocated to those budget headings. An in-depth comparative review has been conducted, using the data obtained from two focus groups that were expressly created for the study, in conjunction with the Second European Survey of Enterprises on New and Emergent Risks (ESENER-2) and its Spanish counterpart (ESENER-2 Spain). The focus groups were formed with agents and entrepreneurs involved in the construction sector, from both the public and the private sector, in order to provide greater impartiality to the resulting data. The principal strategic indicators that served as a guideline for the moderators of the different focus groups were analyzed. The results obtained show great similarity between the data from the focus groups and the data from ESENER-2 and ESENER-2 Spain; which demonstrates the idiosyncrasies that surround this productive sector in the European setting, so badly treated by the economic crisis. All of these points highlight the imperative need to professionalize the construction sector, implementing a “*risk prevention culture*” among all of the agents involved in the constructive-preventive processes that surround construction activities.

## 1. Introduction

Historically, construction activities have always assumed a special role in the stimulation and development of the Spanish economy, mainly due to their capacity to activate other ancillary productive sectors, especially those related to the supply of materials and equipment [1]. Due to the characteristics of the Spanish economy, construction activities are strategic because of the multiplying effect that they have on industrial activity, generating investment and employment in multiple sectors, and boosting the growth of the economy as a whole [2]. Despite the property crisis that occurred between the years 2006 and 2016, the construction sector in Spain has continued to have a significant weight in the economy, greater than in other European countries [3]. With the end of the economic recession, the growth expectations in the construction sector are extremely encouraging, contributing towards the recovery of the Spanish economy as a whole [4].

This invigorating effect on construction activities is an essential factor in the generation of wealth and employment. Indeed, employment in the construction sector in Spain reached its highest level ever in 2004, with a total of 2,455,700 workers [5]. A situation to which small and medium enterprises (SMEs) [6,7] and self-employed workers contributed more than any other segment. The majority of these companies participate directly in projects related to the subcontracting of construction activities in civil works and large-scale building work, in the renovation and restoration of buildings, and in the provision of ancillary services to the construction sector [8,9].

In 2007, with the onset of the economic crisis in Spain, the sector’s accident rates steadily reduced, due principally to the drop in construction activities. This situation led to the collapse of companies and an enormous loss of both direct and indirect employment [10]. In this context of economic downturn, there was a significant setback to gross national production in which the construction sector had always played a leading role [11], which also affected its ancillary industries [12].

Another effect that the economic crisis has had on construction companies has been the inevitable reduction of production costs, in an attempt to improve competitiveness, due essentially to falling revenues in a more competitive scenario. On the one hand, investments in technical, professional and risk prevention training were also affected, with the consequence of heightened risks for worker safety [13].

On the other hand, the basic model of construction companies in Spain, mostly found in the segment of SMEs [14], has no stable organizational structures that guarantee appropriate preventive process management [15,16]. This structural deficit is also a determining factor when the economic crisis threatens the very survival of companies, leaving them with insufficient resources to assume these tasks. In this respect, employee training is a key factor in occupational risk prevention, particularly in the construction sector.

Construction activities, due to the complexity and diversity of tasks that must be carried out either simultaneously or successively, are especially susceptible to accidents at work, despite the efforts of the public authorities to promote safety measures in construction companies through the design and implementation of strategic sectoral plans, allocating significant resources to professionalize workers in the sector and to train them in risk prevention matters [17].

Without specific organizational structures committed to safety management, whether this is because of a lack of resources or due to a lack of “risk prevention culture”, most SMEs outsource these activities to external risk prevention services (EPS) [18].

It is unquestionable that working towards the integration of risk prevention in the management structures of firms is a guarantee of success [19]. A circumstance that, in the case of construction SMEs, would make it possible to control safety in productive processes, integrating direct communication with the workers involved in such processes [20].

Accordingly, it is evident that the “risk prevention culture” of construction companies is a critical factor in the reduction of accident rates [21], and that the preventive management model of the company dictates corporate interest in security and the level of training of its workers [13].

For all of these reasons, it is necessary that the principal agents involved in construction processes, that is companies and their employees, assume their commitments and responsibilities. Company managers must integrate occupational risk prevention in their corporate management systems, and the employees must acquire further skills and expertise in their work through specialist training. It is important to create a “risk prevention culture” amongst all of the agents involved in construction activities.

As a result, the objectives of this research are to refute or reject, using verified data, whether the symptomatology displayed by Europe in respect of the construction sector, is similar to that displayed by Spain, and ultimately establish possible solutions to this issue by analyzing a series of indicators that make this sector the productive sector with the highest rate of accidents at work.

## 2. Preventive Organization and Training: Literature Review

Construction activities are particularly critical from a safety point of view, since the changing environments, subcontracting, and nature of the activities make it impossible to perform systematic process management [22,23,24]. In order to successfully deal with health and safety in construction, the research carried out has highlighted, by way of comparative advantages, management leadership in companies and employee training. Table 1 and Table 2 contain various bibliographical references which have served to disseminate research in this context.

## 3. Methodology

The working methodology used in this research is the prospective-qualitative technique of the focus group. A technique that permits information to be gathered from opinions and experiences contributed by a previously selected group of suitable agents, in a context of debate led by a moderator, in the twin roles of chairperson and timekeeper [56].

The quality of information provided by this technique depends to a large extent on participant experience and commitment; in this particular case, the participants were previously selected in accordance with their involvement in construction processes, such as the case of the administrators and managers of the companies, the workers’ union representatives and those responsible for the administrative bodies in charge of supervising occupational activities and labor relations. The heterogeneity of the group guarantees that different opinions can emerge in the debates, a reflection of the different points of view and appreciations of the same reality as seen from different perspectives [57,58].

Likewise, the focus group technique permits researchers to generate a debate for subsequent analysis that is better oriented to the purpose of the investigation, stimulating the dialogue between agents that hold different points of view on the same matter under discussion [56,59]. If a trusting atmosphere is established, the information contributed to the debate is authentic and of good quality, making it possible to evaluate representative opinions from different perspectives within a short time [60], which means that the results may be managed in a simple manner. 

The results obtained from the focus groups were validated through a comparison with the study conducted by the European Agency for Safety and Health at Work (EU-OSHA) and the results of the “*Second European Survey of Enterprises on New and Emerging Risks*” (ESENER-2) [61], in conjunction with the conclusions from the “*National Survey of the Occupational Risks in Firms*” (ESENER-2 Spain) [62], of the Spanish National Institute for Health and Safety at Work [*Instituto Nacional de Seguridad e Higiene en el Trabajo*] (Spanish acronym: INSHT).

### 3.1. Design of the Focus Groups

Two focus groups were designed for the purposes of this paper (Table 3), formed by agents involved in the management processes of construction sector companies and in the monitoring and control of safety routines in construction works and their productive systems.

### 3.2. Reference Indicators 

In a first phase, the round tables of the different focus groups were designed, and then led and moderated by a member of the research team. The functions of the moderator ranged from the organization and timing of the interventions by the participants, to active encouragement in favor of participation and debate [57,63].

In line with the research objectives, a set of indicators to introduce the topics under debate were established, in accordance with the criteria of the moderator. The indicators used in the discussions of focus group 1 (Table 3) are detailed below:Regulatory framework that operates in the construction sector and characteristics of the sector.Resources allocated to risk prevention training and the quality thereof. Entry barriers due to the singularities of the construction sector.The “risk prevention culture” in the construction sector.

As a corrective measure, no new indicators were identified for the debates established in focus group 2 (Table 3), instead similar indicators were used to those applied by the first Focus Group, giving the participants total freedom to be able to recount their professional and personal experience as construction firms, in relation to the work environment in which their activity took place [64].

## 4. Results

### 4.1. Results of Focus Group 1

In the following sections, Table 4, Table 5, Table 6 and Table 7 provide extracts of the most relevant results referring to the first, second, third, and fourth indicators (Section 3.2), respectively, both from the debate established as a consequence of the focus group, and the conclusions drawn from ESENER-2 Spain [62] and ESENER-2 [63]. 

A clear analogy is appreciated between the data obtained from the discussion groups and the conclusions from ESENER-2 Spain and ESENER-2. This highlights the relevance of the problem, existing in a sector as complex as the construction sector, which cannot and neither should it be treated in the same way as other productive sectors, due to the special casuistry of its operations [19,65]. 

#### 4.1.1. Compliance with the Regulations

With regard to compliance with the regulations in force, the General State Administration, represented by the Spanish Labor and Social Security Inspectorate [*Inspección de Trabajo y Seguridad Social*] (Spanish acronym: ITySS), argued that, in general, there is formal compliance with the regulations (Table 4), because the companies strive to meet all legal requirements in matters of risk prevention. Nevertheless, there is no direct match between this “formal compliance” and the reality of the enforcement processes, where the inspectorate has detected serious shortcomings in matters of risk prevention.

The entrepreneurs from the sector argued that the regulations are very complex and full compliance with them is difficult, because there are many demands and the SMEs do not have sufficient administrative structures for their compliance and management. To compensate for these shortcomings, it is common practice to outsource these activities to the EPS. Additionally, the economic crisis leaves no margin to assume the costs that are generated by management.

The workers’ representatives indicated that greater control should be demanded, because compliance with the obligatory requirements of the regulations is not observed, especially with reference to investment in safety equipment. 

The COAATIE (College of Technical Architects and Building Engineers (Spanish acronym)) representatives indicated that the formal requirements are numerous and it is difficult to control all of the legal requirements, especially with respect to the requirements for subcontracting companies and their workers. As a result, it was observed that the site foremen and the HSCs assumed responsibility for these matters that, at times, they were unable to control or worse still, were not sufficiently well trained to do [66].

The LFC (Labor Foundation Construction) indicated that greater professionalization in construction sector companies is needed, both with respect to the administrators, the technical managers, and the workers themselves [67]. Special attention has been given to the training of manual workers to which large amounts of resources have been allocated [68], but not to managers, administrators, and intermediate staff in the company [69].

The EPS representative admitted that they usually compensate the shortcomings of the SMEs in matters of risk prevention, but that their follow-up of the processes is not continuous. The contracting of their services is occasional and for specific needs, but no monitoring of the construction processes is established throughout the project [70].

#### 4.1.2. Quality of the Training

It is necessary to improve the training of technicians and workers involved in constructive-preventive processes, because they are the real safety managers (Table 5) on construction sites. The ITySS also considers that the training of workers must be more intense and committed [71].

The resources assigned to training in matters of occupational risk prevention are insufficient in the opinion of the workers’ representatives [72]. Moreover, resources would have to be assigned for the improvement of technical knowledge pertaining to constructive activity, by way of continuous and professional training of workers in the sector. The introduction of the professional construction permit [*Tarjeta Profesional de la Construcció*n] (Spanish acronym: TPC) was positively valued.

The ITySS has detected weaknesses in technical and specialized training for workers, because the construction sector has traditionally been a “*refugee sector*” [73], which welcomes workers from all sectors without either technical specialization or specific training in constructive-preventive processes. Hence, the workers are unaware of the dangers that construction activities involve, which is a contributory factor to such a high accident rate [21,74], if compared with other productive sectors.

The representatives of the LFC argued the need to professionalize the construction sector. The TPC is a good instrument so that firms can understand and familiarize themselves with the experiences of construction sector workers, their professional qualifications and technical and preventive training acquired throughout their working careers. In this respect, companies should commit themselves to employing workers that hold the TPC. This permit documents the training that the worker has received in occupational risk prevention and other technical and professional qualifications. 

The business associations argued that most training resources have above all been assigned to manual workers from the construction sector, leaving aside the business entrepreneurs, who in many cases are entrepreneurs with neither training nor specific knowledge of construction processes. Therefore, it would be advisable to organize more specific training programs for entrepreneurs and managers of SMEs [75].

#### 4.1.3. Barriers to Entering the Construction Sector 

In this section, singular aspects were discussed, specific to construction activities, placing special emphasis on the way of organizing places of work and difficulties with their management.

The ITySS indicated that the number of places of work and their distribution are circumstances that do not help when enforcing satisfactory management controls, above all in SMEs [76], which have neither organizational structures nor sufficient human resources and materials (Table 6). 

This way of working is usual in the construction sector, from the entrepreneurs’ point of view, because the main characteristic of construction works is their high temporality, limited to the completion of the construction process. It is evident that the workload obliges contracting in accordance with the production needs, which is why a degree of staff flexibility is necessary. The entrepreneurs certainly recognized that this flexibility would not favor long-term attachment to the firm and empathy towards it.

The workers’ representatives affirmed that the system of contracting has always been that way, but that working conditions have improved with the new Subcontracting Law [77], and job insecurity has decreased, although a lot remains to be done in that field.

Temporary employment gives rise to situations of fluctuating risk [78], for the managers of the HSCs, which directly affects risk prevention management at the places of work. The regulations should give a greater role to the EPS, because their actions are at present limited to formal compliance with risk prevention duties.

The COAATIE representative, the site foremen, and the HSCs have to manage critical situations, with additional difficulties, as well as those pertaining to responsibility for technical implementation processes. The method of contracting by work or service does not facilitate the continuity of workers in the company [79], a circumstance that complicates the management of works in general, and the risk prevention obligations in particular.

Temporary contracts would not have to be a problem, in the opinion of the LFC, if workers were well trained, with regard to technical skills and risk prevention matters [80,81]. The professionalization of the sector would resolve many of the problems of works, especially with regard to accidents at work.

#### 4.1.4. The “Risk Prevention Culture” 

All the participants in the discussion group considered that risk prevention should be a personal and professional value that must be promoted in all available training.

Risk prevention activities constitute the best strategy for achieving more efficient processes, with no accidents and with greater staff satisfaction for the personnel associated with them for the representative of the HSCs (Table 7). Investing in risk prevention is a competitive advantage for construction entrepreneurs, even though investments have fallen over the years of economic crisis.

Companies with good management systems can, in the opinion of the ITySS, apply risk prevention measures in the processes they carry out with greater efficiency. Inspections and site visits may be seen as positive follow-up for compliance with legal requirements, rather than coercive measures. 

The COAATIE representatives pointed to the difficulties that the site foremen and the HSCs have when the construction project is underway, because in addition to technical controls over the works, they have to conduct follow-up of risk prevention obligations, a circumstance that can give rise to significant responsibilities for which they are not sufficiently well trained. It is necessary to assign resources for the ongoing training of Site Foremen and HSCs [71].

The LFC argues that each worker should be responsible for applying the risk prevention knowledge acquired in training procedures [82]. The law determines that risk prevention must be led by the entrepreneur, but it also constitutes a collective duty in which all of the agents that participate in construction site work are involved [83].

The collective of entrepreneurs defended compliance with the risk prevention obligations, such as medical examinations, the company risk prevention plan, the health and safety plan, and training programs. In the case of SMEs, these activities are outsourced as they have neither their own management structures nor risk prevention or staff training resources. They consider that construction companies, despite the crisis, continue investing in risk prevention resources, especially in individual and collective safety equipment. They recognize that the public authorities allocate significant resources to training [75,84], without which the SMEs could not take on these sorts of risk prevention training activities.

### 4.2. Results of Focus Group 2

The discussions between the representatives of the construction companies concerned topics highlighting the principal difficulties that SMEs from the construction sector face in the “*processes of integration of risk prevention management in the general management system of the company*”, in relation to the tasks that they perform, mainly as subcontracting firms (Table 8).

These difficulties can mainly be summarized by saying that SMEs have insufficient means with which to integrate risk prevention management in the general management system of the company. As risk prevention management is a technical-administrative activity, most firms outsource it through an agency or through an External Prevention Service (EPS). They likewise consider the regulation to be excessively complex and difficult to interpret, finding that their organizational structures are insufficient to assimilate so many normative regulations, after having outsourced them. By doing so, the entrepreneur loses the perspective of legal relevancy.

Many of the entrepreneurs from SMEs are workers with manual and technical but no administrative and management training. Hence, more resources have to be allocated to provide management training for these managers.

In the management of the outsourcing processes of construction activities, significant shortcomings are observed in the communication between the contracting and the contracted firm. Clear and defined protocols must be established for the delivery of and access to health and safety plans, incident books, and subcontracting books.

It is necessary to establish similar protocols across the national territory that clearly specify the documentation that has to be delivered by the contracting firm in subcontracting activities, as there is a greater disparity of criteria in this respect, depending on the area in which the works are located.

The entrepreneurs were unanimous when arguing that the requirements of ITySS are oriented more towards formal compliance with the regulatory requirements, than to testing the scope and the quality of their content. The most evident case is that of the health and safety plan, which only requires the approval of the HSCs during the implementation phase, without valuing whether its content has been adjusted to meet the specific aspects and singularities of the work that they will inspect.

Risk prevention activities require important resources. The drop in company revenues because of the effects of the economic crisis, are of no help in this respect. The LFC, the organization dedicated to training and risk prevention in the construction sector, has also reduced its training activity. 

## 5. Discussion

If the results of the first indicator, *regulation and level of compliance in matters of occupational risk management* (Table 4) are analyzed, a clear analogy is observed between what is expressed by the representatives of Business Associations in focus groups 1 and 2, when considering that the regulation of risk prevention matters is “*excessive, disperse, and complex to apply*”. The opinion of entrepreneurs from the sector, focus group 2 (Table 8), on the excessive complexity of the regulations was also corroborated in the ESENER-2 report, where “*for three out of every ten establishments, the circumstances that act as barriers or impediments to risk prevention activity are: the complexity of legal requirements (36.3%), the lack of personnel or the lack of awareness of the employees*” (Table 6). In accordance with the ESENER-2 Spain report, “*bureaucracy and paperwork are excessive*” for 28.7%. In the majority of cases, the regulations are observed, because they are legal obligations, to avoid sanctions in case inspections conducted in Spain (85.4%) and in Europe (77.9%) (Table 4). 

Equally, in accordance with ESENER-2 Spain, “83.1% of the companies inspected considered that their main reason for risk prevention activity was to avoid fines and sanctions from the Inspectorate”, which to a great extent coincided with the message from the ITySS on the coercive nature of the inspections to confirm compliance with risk prevention in companies.

In the case of European companies, the concern about “*prestige and company reputation*” is a reason for their compliance (77%), while the motivation in Spain in that respect is something smaller (65.9%) (Table 4). Likewise, compliance to “*satisfy the expectations of their own workers*” is higher in Europe (78%) than it is in Spain.

Regarding the second indicator, *quality of training*, both the participants of focus group 1, and those of focus group 2, were aware of the need to increase the resources assigned to the risk prevention training of entrepreneurs, technicians, and workers from the construction sector. The economic crisis in Spain has meant fewer resources for training, a circumstance that is also noted in the ESENER-2 Spain survey, where it states that “*14.5% of companies claim to have reduced resources available for risk prevention in the last three years because of economic difficulties*” (Table 5), although there are many companies that state they have a specific budget for risk prevention activities in Spain (68.4%).

The assertion that SMEs have fewer resources to invest in risk prevention, among other reasons because of the structural crisis of the construction sector (focus group 1 and focus group 2), is also corroborated in the ESENER-2 report, where it is affirmed that there are significant differences in matters of preventive training, confirming that “*this training is more frequent in workplaces with over 50 workers than in those with fewer staff*” (Table 5), and that it is higher in Spain than it is in Europe.

Unlike what was stated in focus group 1, where the need for the training of implementation managers and HSCs in project implementation was pointed out, “*82% of the workplaces with a staff of over 19 workers have facilitated training to the Team Managers and Production Managers on how to manage risk prevention*” (Table 8), this activity being especially significant in “*construction, waste management, water and electricity supply, where the highest percentage is observed*”, particular attention being paid to the risk prevention training for site foremen (91.4%). 

The third indicator under study, *barriers to entering the construction sector,* reflects the concerns about risk prevention voiced by entrepreneurs and corroborated by the ITySS itself and COAATIE representatives in focus groups 1 and 2. These concerns are also in harmony with the results of the ESENER-2 report, because it is an entrepreneurial concern of directors and managers of sectors such as “*construction, waste management, water and electricity supply*” in 75% of companies. Nevertheless, in accordance with the ESENER-2 Spain report, the degree of integration of occupational risk prevention is greater “*the larger the size of the workplace*”, standing at 79% in large firms as opposed to 43% in SMEs (Table 5). 

Factors such as high temporality and job insecurity do not encourage the attachment and empathy of workers towards their company, although they may have improved their working conditions following the publication of the Subcontracting Law. High temporary employment does not favor correct risk prevention management in the workplace either.

The “*risk prevention culture*”, the fourth indicator of the study, underlines the imperative need of all the participants to understand risk prevention as a personal and professional value of attachment that has to be promoted at all levels. It is known that investment in risk prevention implies a competitive advantage for entrepreneurs making those investments, but it is directly affected by the lack of resources needed to maintain a stable organizational structure, forcing the majority of construction firms to turn to external risk prevention managers (focus group 1). This is also corroborated in the ESENER-2 Spain report, where it states, for example, “*for the completion of risk evaluations, 78% of workplaces use an external risk prevention service and only 12%, in-house services*”, leaving Spain among the countries that have opted for greater outsourcing of risk prevention management (Table 7). However, the results of the ESENER-2 survey show higher worker involvement in risk prevention assessments (81%), in comparison with Spain (42.8%).

## 6. Conclusions

In general, the results obtained in focus groups 1 and 2 are in line with the conclusions of ESENER-2 [64] and ESENER-2 Spain [65]; a symptom that shows that the construction sector is not a typical productive sector, but rather a sector that features a high level of complexity, which complicates the management and supervision of preventive processes.

In the area of regulatory compliance, the lack of preventive culture has a direct impact on the sector’s accident rate due to significant and evident breaches, making specific actions such as the implementation of awareness/training programs concerning the benefits of integrating risk prevention in all aspects of the company necessary, paying particular attention to the employees who have decision-making responsibility. In order to achieve this, it is necessary that the risk prevention services experts are familiar with the construction process, which implies that these experts should have a professional profile suitable for being an HSC, since it is only when you know the complexity of the sector in depth that you are competent to work in it (Table 4).

The construction sector brings together a group of low-qualified workers, which makes it more difficult to successfully implement preventive measures, and means that all efforts should be focused upon professionalization, with both companies and employees being committed to achieving professional excellence. This field is where training plays a fundamental role, which is why raising the job entry requirements is considered to be a priority, asking for a minimum level of studies and risk prevention training, with a requirement that the latter be continued over time with high-quality training, creating customized training plans for companies, depending on their activity, resources, and size (Table 5).

If we add the lack of personal allegiance to the company’s corporate structure, due to factors such as temporariness and the provisional nature of contractual bonds, to the lack of training, we find ourselves facing great difficulty when it comes to controlling certain situations of risk derived from the lack of roots and empathy with the said company. This means that the management of psychosocial risks represents a fundamental role along with the promotion of campaigns to ensure the correct implementation of related measures, as well as the need to boost staff rotation, with the clear objective of being able to attach the most suitable conditions possible to the job role for the employee occupying the position (Table 6).

In addition to all of the above, it is necessary to encourage greater employee collaboration and participation when assessing the job role, pushing us to ensure that both the presence of the preventive resource and the prevention specialists end up becoming effective (Table 7).

Finally, in order to be able to address all of the above, a “*commitment to risk prevention*” is needed from the boards of directors, as the involvement of directors in risk prevention matters constitutes a key factor for the correct integration and subsequent application of prevention in firms along with the implementation of a “*culture of risk prevention*” among all the agents involved in the constructive-preventive processes that surround construction activities.

## Figures and Tables

**Table 1 ijerph-16-02045-t001:** Bibliographical references concerning research into corporate leadership and the integration of risk prevention in company system.

Organization Indicators	Bibliography
**Engagement of company directors**	Hadakisumo et al., 2017 [25] Jitwasinkul and Hadakisumo, 2011 [26] Kim et al., 2019 [27] Rozenfeld et al., 2010 [22] Windapo, 2013 [28] Jiang et al., 2014 [29]
**Fulfillment of safety regulations**	Cheng et al., 2010 [30] Mohammadi and Tavakolan, 2013 [31]
**Preventive actions in design phase and drafting of project**	Jiang et al., 2015 [29] Esteban et al., 2012 [32] Mohammadi and Tavakolan, 2013 [31]
**Communication and dialogue with employees**	Bavafa et al., 2018 [33] Kim et al., 2019 [27] Marzuki et al., 2012 [34] Newaz eta al., 2019 [35] Seo et al., 2015 [36]
**Leadership of company managers**	Azhar et al., 2009 [37] Grill and Nielsen, 2019 [38] Stiles et al., 2018 [39] Wu et al., 2015 [40] Wu et al., 2017 [41] Yanar et al., 2019 [42] Yulk, 2012 [43] Zang et al., 2018 [44]
**Collaboration and dialogue with subcontracted companies**	Loosemore et al., 2019 [45] Manu et al., 2015 [46] Xu et al., 2019 [47]

**Table 2 ijerph-16-02045-t002:** Bibliographical references concerning employee training in safety and risk prevention.

Training Indicators	Bibliography
**Professionalization of the Construction Sector**	He et al., 2016 [48] Romero et al., 2018 [24] Tie-min, 2008 [49]
**Performance quality and professional competence**	Başağa et al., 2018 [50] Losemore et al., 2019 [45] Marzuki et al., 2012 [34] Xu et al., 2019 [47]
**Safety culture**	Nevaz et al., 2019 [35] Oswald et al., 2018 [51] Seo et al. 2015 [36]
**Learning methodologies**	Shi et al., 2019 [52] Gao et al., 2018 [53] Gao et al., 2019 [54] Poh et al., 2018 [55]

**Table 3 ijerph-16-02045-t003:** Position of the intervening parties/experience in years, duties of the moderator, debate content and duration.

Parties		Moderator	Content/Time		
Provincial Chief of the Labor Inspectorate and Social Security (10 years). Provincial Chief of Health and Safety at Work Service (14 years). Risk Prevention Specialist at the Spanish Confederation of Business Organizations (9 years). Area head at the Labor Foundation Construction (LFC) (22 years). Risk Prevention Specialist at the Construction Labor Foundation (8 years). Director of an ORP company (12 years). Legal advisor of a Professional College of Technical Architects and Building Engineers (Spanish acronym: COAATIE) (21 years). Head of the Risk Prevention Service of a big Spanish company with more than 250 employees (12 years).		Propose the topics to be addressed. Promote participation and desire for discussion. Control the progression of the speeches so that they remain relevant. Provide statistical data from national and European studies concerning the topics to be addressed.	Regulatory framework that operates in the construction sector and characteristics of the sector. Resources allocated to risk prevention training. Complexity of the sector, associated risks and their assessment, preventive structures. Consultation and participation.		
**Focus Group 1: Experts**					
**Total:**	**8**				**150 min**
Company specializing in renovations and restorations (30 years). Company specializing in waterproofing (12 years). Company specializing in new constructions and renovations (15 years). Company specializing in civil and building works (18 years). Company specializing in public works (20 years). Company specializing in the assembly and rental of scaffolding (25 years). Company specializing in electricity (22 years).			Resources allocated to risk prevention training. Difficulties in integrating risk prevention in their companies, in accordance with the reference standards in force. Personal opinion of the content set out in focus group 1.		
**Focus Group 2: Entrepreneurs**					
**Total:**	**7**	**1**			**120 min**

**Table 4 ijerph-16-02045-t004:** Conclusions drawn from focus group 1 and the Second European Survey of Enterprises on New and Emerging Risks (ESENER-2, in both its Spanish and European versions), referring to the first indicator (regulation in matters of occupational risk prevention in the construction sector and level of compliance) used as a guide in discussion groups. Source: [61,62].

Focus Group 1	ESENER-2 Spain	ESENER-2 Europe	Proposal
Formal compliance with the regulations, in other words, compliance is by obligation even though the participants may not believe in them. Excessive yet patchy regulation that is complex to apply. The site foremen and health and safety coordinators (HSC) assume responsibilities for which they are not trained. Greater commitment is needed from workers. Greater support from external risk prevention services (EPS) [*Servicios de Prevención Ajenos*] is necessary.	The regulations are fulfilled because they are considered legal obligations, to avoid possible administrative sanctions (85.4%). The legal requirements are complex and difficult to interpret. Lack of material measures to manage prevention correctly. Lack of resources to carry out frequent visits from the inspectorate. Excessive bureaucracy and procedures for the management of risk prevention in firms.	The regulation is only fulfilled because it is a legal obligation (77.9%), as well as because of the prestige and the reputation of the organization (77.0%). Normative compliance to satisfy the expectations of workers.	Awareness/training programs concerning the benefits of integrating risk prevention in companies. Risk prevention specialists to be better trained in construction, desirable figures being architects, technical architects, engineers, and technical engineers (acceptable professions for the role of HSC).

**Table 5 ijerph-16-02045-t005:** Conclusions drawn from focus group 1 and the Second European Survey of Enterprises on New and Emerging Risks (ESENER-2, in both its Spanish and European versions), referring to the second indicator (quality of training of personnel linked to the construction companies) used as a guide in discussion groups. Source: [61,62].

Focus Group 1	ESENER-2 Spain	ESENER-2 Europe	Proposal
Improve technical training, more professionalization of workers (87.5%). Increase the training of workers in matters of health and risk prevention. It is necessary to professionalize the construction sector (100%). Workers should be required to hold the professional construction permit. More resources must be assigned to the training of managers in construction firms.	A larger percentage is assigned to training in matters of risk prevention in Spain than in Europe. Training in risk prevention for site foremen (91.4%). Training in the management of risk prevention for site foremen (82.0%). Training in risk prevention for delegates and entrepreneurs (96.4%). Training is more common in workplaces with over 50 workers.	Training in management for site foremen (82.0%). Fewer resources are allocated to training in matters of risk prevention. The larger the company, the easier it is to access preventive training.	Increase the job entry requirements. Minimum level of studies and risk prevention training. Recurring training instead of one-off training that lasts for a lifetime and does not need to be updated.

**Table 6 ijerph-16-02045-t006:** Conclusions drawn from focus group 1 and the Second European Survey of Enterprises on New and Emerging Risks (ESENER-2, in both its European and Spanish versions), referring to the third indicator (barriers to entering due to the singularities of the construction sector) used as a guide in the discussion groups. Source: [61,62].

Focus Group 1	ESENER-2 Spain	ESENER-2 Europe	Proposal
Geographic dispersion of places of work. Need for flexibility of staff.	Lack of awareness of employees and to a lesser extent, of the entrepreneurs themselves. Need for flexibility in the sector, because of lack of time and personnel.	Construction is a sector with infinite risks associated with its activity and its processes.	Increased presence and assessment of psychosocial risks in construction works and promotion of campaigns aimed at their correct implementation. Increase staff rotation on lengthy building projects which may result in family-work conflicts.
The worksite is a living organism, in constant change, which implies different risks.			
Lack of empathy and attachment of the workers towards their company.			
Professionalization would solve many current problems in the sector.			

**Table 7 ijerph-16-02045-t007:** Conclusions drawn from focus group 1 and the Second European Survey of Enterprises on New and Emerging Risks (ESENER-2, in both its European and Spanish versions), referring to the fourth indicator (the “risk prevention culture” in the construction sector) used as a guide in the discussion groups. Source: [61,62].

Focus Group 1	ESENER-2 Spain	ESENER-2 Europe	Proposal
Investment in risk prevention implies a competitive advantage for the firm (87.5%). The technicians assume responsibilities for those who are not properly trained (75.0%). The required risk prevention documentation is prepared, such as risk evaluations and health and safety plans. The firms comply with their preventive obligations (100.0%). External prevention services (EPS) are contacted in the majority of cases (100.0%). Risk prevention is the responsibility of all the agents that are involved, not only of a few.	Risk assessments are performed in the construction sector. EPS are contacted in the majority of cases for risk prevention management. Lower involvement of workers in risk assessments (42.8%). There is a s pecific budget for risk prevention (68.4%). The entrepreneurs involve themselves in the risk prevention of the company (75.0%). Personnel discuss risk prevention topics (62.6%). Reduction of resources allocated to risk prevention (18.0%).	Major participation and collaboration of workers in risk assessment (81.0%). The majority of companies periodically complete risk assessments. Major presence of in-house risk prevention services (18.6%), as opposed to EPS. Low participation of risk prevention experts (52.0%). Risk assessments contribute to safety (90.0%).	Awareness campaigns and effective implementation of the figure of the preventive resource in building works, as added value with regard to risk prevention in construction works. Greater presence of building specialists, which implies monthly visits by EPS specialists.

**Table 8 ijerph-16-02045-t008:** Conclusions drawn from focus group 2 and the Second European Survey of Enterprises on New and Emerging Risks (ESENER-2, in both its Spanish and European versions), referring to risk prevention management in construction sector companies. Source: [61,62].

Focus Group 2	ESENER-2 Spain	ESENER-2 Europe	Proposal
Excessive, very dispersed regulations that are difficult to interpret. Lack of measures for the correct integration of risk prevention in the management of the company. More resources need to be dedicated to company training managers (71.4%). Need to increase the resources in risk prevention training. Broader and better communication must be established between contractors and subcontractors (85.7%). Unification of the management protocols with the subcontractors is imperative. The current regulations are oriented towards formal compliance with risk obligations (100%).	Firms make use of an external prevention service (EPS) for the organization of risk prevention. Lack of measures for the correct integration of risk prevention in the management of the company. Serious difficulties are found when applying the regulations. 82% of places of work with a staff of over 19 workers provide training to team and production managers on how to manage prevention. Lack of “risk prevention commitment” among the entrepreneurs and the workers on the integration of risk prevention in the company. The involvement of directors in risk prevention constitutes a key factor for boosting risk prevention in places of work.	Personnel are more involved in risk assessments and in risk prevention activities. Companies are concerned about their image and reputation. 90% of firms make a document available to workers that explains the responsibilities and the procedures in Health and Safety matters. A specific annual budget is set for the measures and health and safety equipment (41%). The participation of directors in health and safety is an essential factor for the application of measures in that field.	Simplify preventive documentation, even replacing it with sheets that set out the construction processes and list the potential risks. Supplement the documents with informative chats and regular training courses that reinforce prevention.

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
