# Peer review of "Mind the Gap: Professionalization is the Key to Strengthening Safety and Leadership in the Construction Sector"

_ijerph, 2019, doi:10.3390/ijerph16112045_

Round 1

Reviewer 1 Report

The authors analyze the reality of risk-prevention in construction firms in Spain by the analysis of training, management, and risk-prevention. Two Focus Groups meetings are conducted. The results obtained show the great similarity between the data from the Focus Groups and the data from survey data, which demonstrates the idiosyncrasies that surround this productive sector in the European setting, so poorly treated by the economic crisis. All these points highlight the imperative need to professionalize the Construction Sector.

1.. the language of the paper needs to be proofread by a professional editor.

2.. I was wondering whether the results are applicable to countries other than Spain. It seems to me that the results are only applicable to Spain. Then, I do not see much academic value in it.

3.. An expert focus group was formed and an entrepreneur group was formed. Please provide the names, posts, emails, telephone numbers of the 15 people in the response report.

4.. Section 3.1.3 mentioned a number of barriers to entering the construction industry. Are there any strategies to overcome the barriers?

5.. It is mentioned that “Investing in risk-prevention is a competitive advantage for construction entrepreneurs, even though investments have fallen over the years of economic crisis.” But, how much investment in risk-prevention is recommended? I am sure it is not 90% of the revenue.Without mentioning a detailed percentage, it is meaningless pointing out that investing in risk-prevention is good.

6.. Figure 3 does not look like an academic paper. It does not look professional.

7.. Figure 1 is poorly drawn. Should not ‘10000000’ be ‘10,000,000’?

8.. The authors must associate/compare the findings with the results in other countries to identify the differences between the situations in the Spanish construction industry and other countries’ construction industry.

Author Response

The authors are very grateful to the reviewer and to the editor for their suggestions to improve the quality of the article.

Reviewer 01:

1.     The language of the paper needs to be proofread by a professional editor.

Dear Reviewer, following your recommendation, a full review of the document, including its grammar and syntax, has been performed by a professional translator and a certificate demonstrating their professional qualification is attached.

2.     I was wondering whether the results are applicable to countries other than Spain. It seems to me that the results are only applicable to Spain. Then, I do not see much academic value in it.

As stated in the Second European Survey of Enterprises on New and Emerging Risks (ESENER-2), as well as in the different versions produced by European Union member states, although it is true that issues in the construction sector transcend barriers, it is possible to observe characteristics which are typical of the building style of each member state. However, the root causes are entrenched in all countries, displaying the same symptomatology and high numbers of accidents at work.

As such, the issues directly affecting Spain can, to a large extent, be extrapolated to the other EU member states. The section concerning the review of the literature demonstrates this, the same symptomatology being displayed in studies carried out in different countries.

3.     An expert focus group was formed and an entrepreneur group was formed. Please provide the names, posts, emails, telephone numbers of the 15 people in the response report.

Following your recommendations, a table has been added to the text which sets out the position and experience in years of the different members of both Focus Groups. The duty of confidentiality that the authors have with regard to the participants’ personal and professional information stems from the provisions of Organic Law 3/2018, of 5 December, concerning the Protection of Personal Information and Guarantee of Digital Rights, the transposition to Spanish Law of European Parliament and Council Regulation (EU) 2016/679, of 27 April 2016, in relation to the protection of natural persons, with respect to the handling of their personal information and the free circulation of this information. As such, the information that you request may only be provided with the express consent of the participants themselves, and we are consequently neither empowered nor entitled to provide sensitive or personal information concerning the participants.

4.     Section 3.1.3 mentioned a number of barriers to entering the construction industry. Are there any strategies to overcome the barriers?

In accordance with your request, it was decided that a column would be added for proposals for improvement in all of the tables included in the document, by way of a possible strategy for overcoming each of the obstacles analyzed.

5.      It is mentioned that “Investing in risk-prevention is a competitive advantage for construction entrepreneurs, even though investments have fallen over the years of economic crisis.” But, how much investment in risk-prevention is recommended? I am sure it is not 90% of the revenue. Without mentioning a detailed percentage, it is meaningless pointing out that investing in risk-prevention is good.

It has been decided to include a section featuring a review of the literature, containing several studies about the investment in safety. Whilst it is true that there is no established percentage which guarantees total safety, it may be concluded that safety in the workplace improves when a greater investment is made in risk prevention and safety training in companies, reducing the accident rate.

6.     Figure 3 does not look like an academic paper. It does not look professional.

Indeed, and following your instructions this figure, which does not look professional, has been removed and replaced by a table which summarizes the data for both Focus Groups.

7.      Figure 1 is poorly drawn. Should not ‘10000000’ be ‘10,000,000’?

Indeed, this error was corrected in an earlier draft, although it was decided to remove this figure from the corrected version of the document that was submitted, since following the reviewers’ recommendations, the introduction to the document was reviewed to give it a more sharpened focus, directed at the research carried out.

8.     The authors must associate/compare the findings with the results in other countries to identify the differences between the situations in the Spanish construction industry and other countries’ construction industry.

The tables in the document set out a comparison/association between the data obtained in Europe and Spain, in conjunction with the data provided by the Focus Group members. Following your advice, this comparison is highlighted in the Conclusions section of the document.

Reviewer 2 Report

The authors conduct a comparative review to analyze the reality of risk-prevention in Construction Sector firms in Spain. The significant of the topic itself is limited and the manuscript has a number of issues.
1 The participants are divided into two Focus Groups, The Basis for grouping and the detailed messages of the participants are not given. For example
experience ,which impact the result largely.

2 In line 125, The Focus Group technique can hold different points of view on the same matter, but from the perspective of results, there is no diversity of conclusions.

3 The method presented in the paper does not introduce any novelty

4 Because the limitation of the data and methods, the results are relatively simple.

Author Response

The authors are very grateful to the reviewer and to the editor for their The authors are very grateful to the reviewer and to the editor for their suggestions to improve the quality of the article.

Reviewer 02:

The authors conduct a comparative review to analyze the reality of risk-prevention in Construction Sector firms in Spain. The significant of the topic itself is limited and the manuscript has a number of issues.

1.   The participants are divided into two Focus Groups, The Basis for grouping and the detailed messages of the participants are not given. For example ‘experience’ ,which impact the result largely.

Dear Reviewer, following your recommendations, a table has been added to the text which sets out the role performed and professional experience in years of the different members of both Focus Groups.

2.   In line 125, The Focus Group technique can hold different points of view on the same matter, but from the perspective of results, there is no diversity of conclusions.

In accordance with your evaluation, it was decided that the conclusions section should be reworded in order to emphasize the different points of view of the Focus Group.

3.   The method presented in the paper does not introduce any novelty.

Following your recommendations, it was decided that a column would be added for proposals for improvement in all of the tables included in the document, by way of a possible strategy for overcoming each of the obstacles analyzed.

Despite not being an innovative method in the strict sense, its value as a scientific study is relevant, since it analyzes a complex reality which, according to the bibliography provided, is also common in other countries. Furthermore, the study carried out is founded on solid scientific rigor, since valid result gathering and comparison techniques have been used, by means of the methodological use of combined techniques.

4.   Because the limitation of the data and methods, the results are relatively simple.

We agree with the Reviewer. Many of the results are probably obvious for the engineers who work on building projects every day. As is explained in the new bibliography provided in the revised document, the construction sector is complex, and both safety management and employee training are two critical aspects which explain many of the accidents on building sites. The commitment of company managers and employees is a key factor in preventing accidents. The results may be simple, but their influence is significant.

According to the literature consulted, these are two factors which also concern researchers in other countries. In essence, problems in building activities are common to all countries, irrespective of their economic strategies: temporariness, lack of training, lack of corporate commitment, excessive outsourcing (subcontracting) etc.

This is the true reality and as researchers we have the obligation to highlight these circumstances, offering improvements in order to control and suitably manage them.

Reviewer 3 Report

-       There are several grammatical and syntax errors that need to be corrected.

-       The first paragraph (lines 35-37) in page 1,is very short.

-       The Introduction section is very lengthy. It has two figures and one table. The authors should move the two figures and table to the literature review section.

-       The authors should develop a literature review section.

-       The objectives of the study are not well stated in the Introduction section.

-       The authors should describe the process of developing the 4 indicators (lines 182-186)

-       The authors should develop a series of recommendations to overcome the challenges and facilitate strengthening safety and leadership in the construction sector.

-       The Conclusions section needs to be more elaborate to report on the findings of the study.

Author Response

The authors are very grateful to the reviewer and to the editor for their suggestions to improve the quality of the article.

Reviewer 03:

1.   There are several grammatical and syntax errors that need to be corrected

Dear Reviewer, following your recommendation, a full review of the document, including its grammar and syntax, has been performed by a professional translator and a certificate demonstrating their professional qualification is attached.

2.    The first paragraph (lines 35-37) in page 1 is very short.

Further comments have been added concerning the strategic importance of the construction sector in the Spanish economy and future expectations. New bibliographical references have been included to justify the comments added. Despite being a short paragraph, it is of extreme importance, since it briefly and concisely summarizes the evolution of construction and its role in the Spanish economy. The bibliography provided supports the comments made.

3.   The Introduction section is very lengthy. It has two figures and one table. The authors should move the two figures and table to the literature review section.

In accordance with the Reviewer’s recommendations, the said section of the Introduction has been rewritten and the bibliographical references have been reorganized. Finally, it was decided that the tables and figures making up the aforementioned section of the document should be removed.

4.   The authors should develop a literature review section

In accordance with your evaluation, a new section has been included featuring two tables (Table 1 and Table 2) with bibliographical references concerning the organization of risk prevention and employee training in construction companies. As may be observed, they are publications by other authors who have carried out research that looks closely at the indicators which serve as reference indicators for the research detailed in the present document.

5.   The objectives of the study are not well stated in the Introduction section.

As a result of your evaluation it was decided that the objectives of the study should be rewritten in the introduction of the document.

6.   The authors should describe the process of developing the 4 indicators (lines 182 186)

Following your evaluation, it was decided that the aforementioned section of the document should be rewritten, focusing on the indicators that are subsequently to be analyzed in the document.

7.   The authors should develop a series of recommendations to overcome the challenges and facilitate strengthening safety and leadership in the construction sector.

In accordance with your request, a new column was added containing proposals for improvement in all of the tables included in the document, by way of a possible strategy for overcoming the challenges associated with each of the obstacles.

8.   The Conclusions section needs to be more elaborate to report on the findings of the study.

Following your recommendations, the Conclusions section has been reworded, the new version highlighting the research indicators and conclusions extracted.

Round 2

Reviewer 1 Report

The paper is improved over the previous version. My comments are well addressed.

Reviewer 3 Report

All comments were answered. I recommend accepting the manuscript.